# Cytomegalovirus Viral Load in Transplanted Patients Using the NeuMoDx™ (Qiagen) Automated System: A 1-Month Experience Feedback

**DOI:** 10.3390/v13081619

**Published:** 2021-08-16

**Authors:** Léa Luciani, Denis Mongin, Laetitia Ninove, Antoine Nougairède, Kevin Bardy, Céline Gazin, Remi N. Charrel, Christine Zandotti

**Affiliations:** 1Unité des Virus Émergents (UVE), Aix-Marseille Université, IRD 190-Inserm 1207, 13005 Marseille, France; laetitia.ninove@univ-amu.fr (L.N.); antoine.nougairede@univ-amu.fr (A.N.); remi.charrel@univ-amu.fr (R.N.C.); christine.zandotti@ap-hm.fr (C.Z.); 2Laboratoire de Microbiologie, Assistance Publique-Hôpitaux de Marseille, IHU Méditerranée Infection, 13005 Marseille, France; kevin.bardy@ap-hm.fr (K.B.); celine.gazin@ap-hm.fr (C.G.); 3Faculté de Médecine, Université de Genève, 1205 Genève, Switzerland; denis.mongin@unige.ch; 4Comité de Lutte Contre les Infections Nosocomiales, Hôpitaux Universitaires de Marseille (AP-HM), 13005 Marseille, France

**Keywords:** cytomegalovirus, CMV, viral load, NeuMoDx, R-GENE

## Abstract

Cytomegalovirus (CMV) reactivations represent a significant morbidity and mortality problem in transplant patients. Reliable and rapid measurement of CMV viral load is a key issue for optimal patient management. We report here the evaluation of NeuMoDx™ (Qiagen) in a routine hospital setting (University Hospitals of Marseille, France) in comparison with our classical reference technique R-GENE. During one month, 719 CMV viral loads from 507 patients were measured in parallel in both techniques. Using the ROC (receiver operating characteristic) curve and our biological experience we suggest that values <52 IU/mL (geometric mean) correspond to negative samples, values >140 IU/mL (Fowlkes–Mallows index) correspond to quantifiable positive results and values ranging from 52 to 140 IU/mL represent non-quantifiable positive results. Follow-up of 15 transplant patients who developed CMV reactivation during the study showed that NeuMoDx™ provided higher viral load measurement during the first two weeks of follow-up for three patients. These important intra-individual variations resulted in a significant median increase considering the whole data set (6.7 points of difference expressed as a percentage of the initial viral load). However, no difference between the two techniques was noticeable after two weeks of treatment. Subsequent to this first study we conclude that NeuMoDx™, used with optimized logistics and an adapted threshold, allows a rapid CMV viral load measurement and that its use does not lead to any difference in patient management compared to the reference technique R-GENE^®^.

## 1. Introduction

Cytomegalovirus (CMV) is a common double-stranded DNA virus of the *Herpesviridae* family that infects people of all ages. It can remain latent in bone marrow progenitors and monocytes, triggering reactivations throughout human life [1]. CMV is currently a major health problem, in particular for patients with impaired cell-mediated immunity [2], such as solid organ transplant recipients, hematopoietic cell transplant recipients [3], HIV-infected patients [4], and patients treated with immunomodulating drugs [5]. For those immunocompromised patients, CMV infection or reactivation cause several morbidities, including gastrointestinal tract disease, pneumonia, hepatitis, and encephalitis, as well as increasing the chance of organ rejection and opportunistic infections [6]. Hence, CMV viral load monitoring is recommended. The medical follow-up of transplant recipients in France is mainly carried out in “day hospitals”. Upon admission, as part of the biological assessment, an EDTA (ethylenediaminetetraacetique acid) blood sample is collected for CMV viral load detection and results are expected by noon in order to allow adaptation of the treatment or management if necessary; briefly, a negative result will allow hospital discharge, whereas a CMV-positive result could lead to immediate treatment monitored by sequential CMV testing. Therefore, a rapid method for CMV viral load measurement suitable with routine laboratory diagnosis is needed. NeuMoDx™ (Qiagen) is an integrated “random-access” analyzer that provides results within 4 h. It was implanted in a routine laboratory and specimens were processed for CMV DNA quantification in parallel to the reference R-GENE^®^ system. First, we compared the two techniques using a Bland–Altman [7] plot and the receiver operating characteristic (ROC) curve [8] in order to evaluate the performance of the NeuMoDx™ compared to the R-GENE^®^ technique. Second, we compared both techniques prospectively on transplanted patients.

## 2. Materials and Methods

### 2.1. Study Design

A prospective study testing all EDTA blood samples collected in the microbiology laboratory of the University Hospitals of Marseille (UHM), France, from 1 June 2020 to 3 July 2020 with CMV viral load prescription was carried out. Lack of volume for full testing was the only exclusion criterion. Samples were analyzed in parallel using R-GENE^®^ (Biomerieux diagnostics) as the standard routine method and the NeuMoDx™ analyzer. This study was approved by the ethics committee of UHM under the reference 2021-94.

### 2.2. R-GENE^®^ Method

Total nucleic acids were purified from 200 µL of EDTA whole blood using the EZ1 DNA Tissue Kit on an EZ1 Advanced XL instrument (Qiagen, Courtaboeuf, France). Amplification was then performed using the quantitative real-time PCR CMV R-GENE^®^ kit (bioMérieux, Marcy l’étoile, France) targeting the UL83 protein gene in the CMV genome [9] (LightCycler 480 from Roche Life Science). According to manufacturer, the detection limit of the CMV R-GENE^®^ kit is 446 copies/mL and the quantification limit is 1000 copies/mL.

### 2.3. NeuMoDx™ Method

One milliliter of EDTA plasma was loaded onto a NeuMoDx™ CMV Quant Test Strip (Qiagen) in a NeuMoDx™ 96 Molecular Systems analyzer for both nucleic acid extraction and quantitative PCR targeting the UL54 and UL71 genes. The result is positive even if only one of the two target genes is amplified. According to the manufacturer, the detection limit is at 17 UI/mL and the quantification limit is at 20 UI/mL.

### 2.4. Data Analysis

All analyses were performed using R 3.6.3. [10]. Copies/mL to UI/mL conversions were performed using a 1.62 factor calculated as previously described by the WHO method [11]. NeuMoDx™ and R-GENE^®^ measurements were compared using a Bland–Altman plot [7] on the log10 of the measurements [11]. To compare the kinetics of viral load, each measurement performed at a follow-up day (FU) was expressed as a percentage of the viral load in the first sample. Comparison between the kinetics of viral load produced by the two methods was performed using a paired Wilcoxon test.

## 3. Results

During this study, 719 samples from 507 patients were analyzed using both methods. CMV-DNA detection was undetermined in 19 samples (2.64%) using the NeuMoDx™ method, whereas none were undetermined with the R-GENE^®^ method.

### 3.1. NeuMoDx™ and R-GENE^®^ Comparison

To compare the two methods, we calculated the log10 of all non-zero measurements expressed in UI/mL for both methods and plotted the corresponding Bland–Altman plot [7] (*n* = 76) (Figure 1a). The mean of the ensemble of the measurement differences was close to 0, and almost all measurements differences were in their mean range +/− 1.96 SD, indicating a good agreement between the two methods. The 107 points forming a diagonal in the upper part of the graph correspond to the situation where NeuMoDx™ yields non-null measurements while R-GENE^®^ measurements are null. The eight points aligning in a diagonal in the lower part of the graph correspond to non-null measurements for R-GENE^®^ while NeuMoDx™ measurements are null. These are the only measurements exceeding the standard error range, indicating that R-GENE^®^ sometimes provides excessive values. The cloud of points on the right part of the graph (for mean values above 100, that is, for an abscise above 2) is the ensemble of measurements where both techniques were positive. We observe in this case that NeuMoDx™ tended to provide lower values than GENE^®^, with an absolute bias decreasing with the increase of the measurement values. The Spearman rank correlation of the log10 measurements for positive samples in both techniques was 0.79 (*p* < 0.001).

The excess of situations where NeuMoDx™ measurements were non-null and R-GENE^®^ ones were null tends to indicate that the NeuMoDx™ quantification limit of 20 IU/mL is too low. CMV viral load is subjected to physiological variations without biological or clinical relevance for patient management. However, such variations, if detected with a test with exceeding sensitivity, result in worrying situations for both patients and physicians. Accordingly, we performed an ROC curve analysis using the R-GENE^®^ as a reference technique (Figure 1b and Table 1). Optimizing the geometric mean of the specificity and sensitivity led to a threshold at 52 IU/mL, while the optimization of the geometric mean of the sensitivity and the positive predictive rate (Fowlkes–Mallows index [12]) yielded a threshold at 140 IU/L.

### 3.2. Sequential Testing of Transplanted Patients for Therapy Monitoring

The results observed with the two techniques were converted as a percentage of the initial measured viral load (%CMV_0_). Of the 507 patients tested, 15 were recipients of transplanted organs (kidney (*n* = 12), liver (*n* = 2), and lung (*n* = 1)) with a CMV reactivation and for whom at least two samples were collected. All patients received anti-CMV treatment (ganciclovir or valganciclovir) from D0 except for patient 9, whose treatment was initiated at D7. Viral load kinetics of these 15 patients are shown in Figure 2 (one graph per patient). NeuMoDx™ tended to estimate high viral loads, especially during the first days of follow-up. The ensemble of the paired differences between NeuMoDx™ and R-Gene^®^ %CMV_0_ had a median (IQR) of 0.26 [0.00, 8.99] points, and was significantly different from 0 (Wilcoxon test, *p* < 0.001). Again, this indicates that NeuMoDx™ provided a significantly higher measurement of the viral load when expressed as a percentage of the first viral load, and that the median difference was of 0.3 points. When inspecting these measurement differences within four time periods of follow-up (1–7 days, 8–14 days, 15–20 days, and 31–60 days), the median difference was higher during the first two weeks (6.5 and 6.9 point difference of %CMV_0_), whereas it was no more noticeable after the second week (Figure 3 and Table 2).

## 4. Discussion

### 4.1. Strength and Limitations

Our study is technically robust; we have tested a large number of samples with two techniques in parallel under real-life conditions (analysis performed on the day of sampling without prior storage). Furthermore, this is the first independent evaluation of the NeuMoDx™ CMV assay. However, our study also has limitations, such as (i) the unbalanced proportion of negative results in both techniques (89%) compared to positive results (11%), which can preclude the threshold determination for the ROC curve, and (ii) the limited number of patients in the follow-up study.

### 4.2. Comparing the Two Techniques

The comparative analysis of results observed by the NeuMoDx™ and R-Gene^®^ systems’ CMV DNA detection and quantitation showed differences in analytical performance and usability. Regarding analytical performances, the NeuMoDx™ has a detection limit that is lower than the R-Gene^®^ assay. In accordance with the ROC curve and our routine experience with the NeuMoDx™ subsequent to this study, we propose that (i) values <52 IU/mL correspond to negative samples, (ii) values ranging from 52 to 140 IU/mL represent non-quantifiable positive results, and (iii) values >140 IU/mL correspond to quantifiable positive results. We therefore assume that is a proper compromise to mitigate the on–off effects that likely arise due to physiological variations while maintaining improved sensitivity compared to R-GENE^®^. Additional studies are needed to measure the medical importance of an increased sensitivity during patient follow-up with a special interest in the early detection of relapses.

In the clinical management of CMV reactivation in transplant recipients, initiation of treatment is driven by a high viral load and/or disabling symptoms [13,14]. Then, treatment efficacy is reflected by the decrease/suppression of detectable viral load. The effectiveness of the treatment is then monitored and guaranteed by the decrease of the viral load. If after two weeks the viral load has not decreased significantly, resistance of the CMV to the treatment used must be considered [15]. Here, the distribution of the differences between the two methods is right-skewed, signaling that in a few cases, NeuMoDx™ yielded a higher viral load evolution during the first days of follow-up. Larger studies are needed to explore these intra-individual variations. However, our study illustrates that although NeuMoDx™ can in some cases yield a slower decrease of the viral load during the first two weeks, the difference between R-GENE^®^ and NeuMoDx™ measurement of the viral load evolution is low for most of patients and become null after 15 days. As a consequence, the management of treatment against CMV infection in transplanted patients would be similar regardless of the assay used for CMV DNA detection and viral load quantitation.

### 4.3. Adapting the NeuMoDx™ to Hospital Routine

We have identified two problems with NeuMoDx™: (i) the 1 mL volume of plasma requires using a specific tube for collection and this is problematic for newborn patients; (ii) it is validated for plasma only, which requires maintaining another technique in the laboratory for testing other types of samples (urine, respiratory samples, conjunctival samples, biopsies, amniotic fluids, etc.). Moreover, during this study the rate of undetermined samples using NeuMoDx™ was 2.64%. This was mainly due to technical errors during the initial use of the machine. However, with hindsight, we have noticed that the rate of undetermined samples varies depending on the patient population. For transplant patients it is now less than 1%, but for intensive care patients it is close to 10%. We assume that samples from this population are more likely to contain PCR inhibitors such as heparin (further investigations are required). Although dilution of the sample to reduce PCR inhibitors may work for some samples, a second PCR technique such as R-GENE^®^ is still needed. It is also noted that the UL54 gene, which is one of the targets of amplification by NeuMoDx™, is a gene that may have mutations that can confer resistance to ganciclovir, cidofovir, and/or foscavir [16]. As the exact sequences of the primers used by NeuMoDx™ are patent-protected we do not know whether they target a conserved sequence or whether resistance mutations may impact the viral load measurement. However, the dual targeting of the machine limits the risk of false negatives in this case.

The principal advantage of NeuMoDx™ resides in its “random access” operating mode, which directly uses the primary tube and provides an interpreted result within less than 4 h based on a frequency of 40 to 60 samplings per day. The strict technical time of execution of the machine for a sample is even shorter: 1 h and 10 min (extraction and PCR) to which it is necessary to add 10 min of preliminary centrifugation. Hence, samples can be processed as they are received in the laboratory without the need to organize for series that delay the time to result to 12–24 h (i.e., the R-GENE^®^ technique). In a day hospital setting, this advantage is immense, since the result is provided to the medical staff during duty hours.

## Figures and Tables

**Figure 1 viruses-13-01619-f001:**
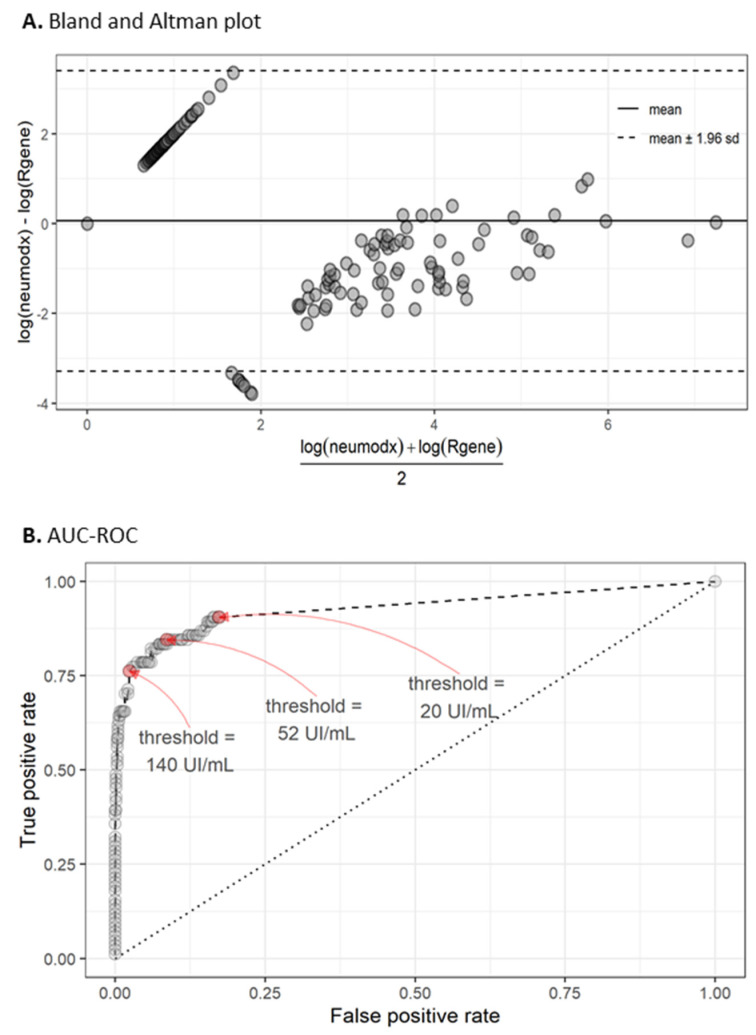
Comparison of the two methods of measuring CMV viral load. (**A**) A Bland–Altman plot (log10 values in IU/mL) showing that all non-zero measurements are coordinated. This graph also highlights the large number of null R-GENE^®^ and non-null NeuMoDx™ measurements (diagonal dots at top left) due to the 50-fold lower quantification threshold of NeuMoDx™ compared to R-GENE^®^. (**B**) AUC-ROC (air under curve-receiver operating characteristic) showing the performance (sensitivity and specificity) of the NeuMoDx™ using R-GENE^®^ as the reference method. A 140 UI/mL threshold corresponds to the optimization of the geometric mean of the sensitivity and the positive predictive rate (Fowlkes–Mallows index). A 52 UI/mL threshold corresponds to the optimization of the geometric mean of the specificity and sensitivity (G-mean). A 20 UI/mL is the commercial threshold.

**Figure 2 viruses-13-01619-f002:**
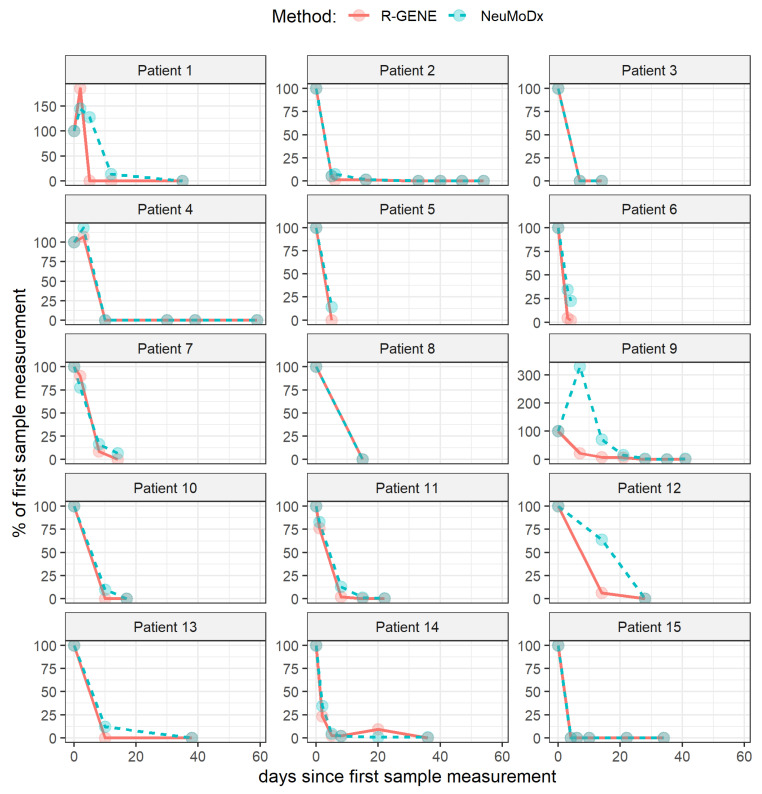
Evolution of CMV viral load kinetics for 15 solid organ recipient patients presented with CMV reactivation during our study. Data obtained from the two techniques were converted as a percentage of the initial measured viral load (%CMV_0_). All patients received anti-CMV treatment from D0, except for patient 9, whose treatment was initiated at D7.

**Figure 3 viruses-13-01619-f003:**
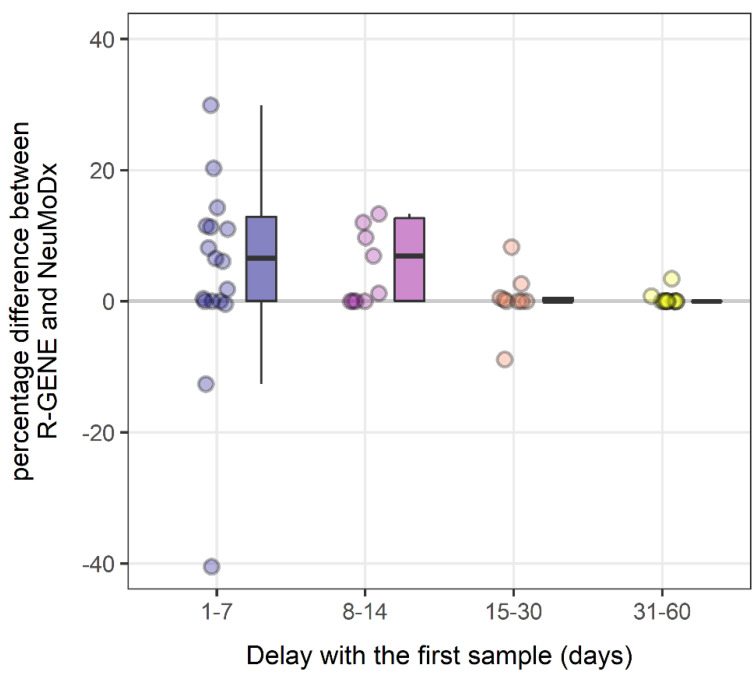
Differences of the percentage of initial viral load measured by NeuMoDx™ and R-GENE^®^ during 4 follow-up periods (1–7 days, 8–14 days, 15–30 days and 31–60 days). The y scale is truncated to an absolute difference of 40 points, for readability.

**Table 1 viruses-13-01619-t001:** Details of the ROC curve thresholds given by the different types of scores. G-mean stands for geometric mean and FM for the Fowlkes–Mallows index. These two indexes can yield a score between 0 and 1. The optimum values obtained are indicated in the score value column, together with the corresponding threshold detection for the NeuMoDx™ measurements, the number of true positives, true negatives, false positives, false negatives, and the rate of false positives (FP rate) and of true positives (TP rate).

Index	Score Value	Threshold (UI/mL)	True Positives	True Negatives	False Positives	False Negatives	FP Rate (%)	TP Rate (%)
G-mean	0.879	52	71	563	53	13	8.6	84.5
FM	0.786	140	64	601	15	20	2.4	76.2

**Table 2 viruses-13-01619-t002:** Wilcoxon signed rank test testing whether the difference between NeuMoDx™ and R-GENE^®^ measurements is different from 0, for the whole follow-up period and for 4 different follow-up periods.

Variable	Overall	Days 1–7	Days 8–14	Days 15–30	Days 31–60
Number of measurements	51	19	11	9	12
R-GENE NeuMoDx median (IQR) difference	0,3 [0.0, 9.0]	6.5 [0.0, 12.9]	6.9 [0.0, 12.7]	0.0 [0.0, 0.5]	0.0 [0.0, 0.0]
Difference different from 0? (*p* value)	Yes (0.0003)	Yes (0.032)	Yes (0.022)	No (0.59)	No (0.21)

## Data Availability

The data that support the findings of this study are available from the corresponding author upon reasonable request.

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
