# Peer review of "Cytomegalovirus Viral Load in Transplanted Patients Using the NeuMoDx™ (Qiagen) Automated System: A 1-Month Experience Feedback"

_viruses, 2021, doi:10.3390/v13081619_

Round 1

Reviewer 1 Report

The manuscript by Luciani and colleagues assesses the NeuMoDx (Qiagen) automated system to quantify the cytomegalovirus  load in transplant patients. Although interesting the manuscript has to be improved as follows:

Major points:

  • The reactivation of HCMV infection has been monitored only in solid organ transplant recipients. This limits the scope of the study. In fact bone marrow transplant recipients are often infected by CMV which can result in a poor outcome in some of them. The addition of biological samples isolated from bone marrow transplant recipients in the study is critical to give a more complete view on the use of the NeuMoDx automated system on a daily basis in a university hospital.
  • It is specified that “2.64% samples were undetermined using the NeuMoDx method, more than with the R-GENE assay”. This could impact the results obtained on a daily basis and can mean that if these samples are undetermined with the NeuMoDx method, the hospital needs to have also a R-GENE assay to complete the measurement of the CMV viral load. Please confirm and discuss it.
  • The NeuMoDx method is based on the quantification of CMV UL54 gene. Since UL54 mutation confers high level of ganciclovir resistance, how can the biologist be sure that the UL54 gene is not mutated in the HCMV strain analyzed. In other words if using the NeuMoDx test the result is negative or undetermined could it be due to a mutation in the UL54 gene? To note that UL54 mutation also confers cross-resistance to cidofovir and foscarnet. Please discuss.

Minor points:

  • HCMV is latent in bone marrow progenitors, but also in monocytes. This should be corrected accordingly in the text.
  • The English has to be checked.

Author Response

The manuscript by Luciani and colleagues assesses the NeuMoDx (Qiagen) automated system to quantify the cytomegalovirus load in transplant patients. Although interesting the manuscript has to be improved as follows:

Major points:

The reactivation of HCMV infection has been monitored only in solid organ transplant recipients. This limits the scope of the study. In fact bone marrow transplant recipients are often infected by CMV which can result in a poor outcome in some of them. The addition of biological samples isolated from bone marrow transplant recipients in the study is critical to give a more complete view on the use of the NeuMoDx automated system on a daily basis in a university hospital.

Thank you for this interesting comment. We are indeed aware that the absence of marrow transplant recipients reduces the diversity of the study. However, in Marseille, adult marrow transplant patients are not followed in our hospital but in a specialised centre which has its own medical analysis laboratory. We have a few (n<5) child marrow transplant recipients among the 507 patients screened during the study period, but they were all negative in both techniques and none of them presented CMV reactivation. We could not do a retrospective inclusion either because for children with marrow transplantation clinicians prefer an R-GENE® assay systematically in order to be harmonized with different French national follow-up cohorts. We therefore only stored whole blood in the biobank and not plasma.

It is specified that “2.64% samples were undetermined using the NeuMoDx method, more than with the R-GENE assay”. This could impact the results obtained on a daily basis and can mean that if these samples are undetermined with the NeuMoDx method, the hospital needs to have also a R-GENE assay to complete the measurement of the CMV viral load. Please confirm and discuss it.

Indeed, the undetermined rate in NeuMoDx is one of the reasons why a second CMV assay technique should be retained. During this study the indeterminate rate of 2.64% was mainly due to the difficulty of handling the new machine. With the benefit of hindsight and the training of technical staff this rate is now much lower in the transplant patient population. We have added a few lines to the discussion to further explore the topic of undetermined samples: “During this study the rate of undetermined samples in NeuMoDx™ was 2.64%. This was mainly due to technical errors during the initial use of the machine. However, with hindsight, we have noticed that the rate of undetermined samples varies depending on the patient population. For transplant patients it is now less than 1%, but for intensive care patients it is close to 10%. We assume that samples from this population are more likely to contain PCR inhibitors such as heparin (further investigations are required). Although dilution of the sample to reduce PCR inhibitors may work for some samples, a second PCR technique such as R-GENE® is still needed.” Lines 204-212

The NeuMoDx method is based on the quantification of CMV UL54 gene. Since UL54 mutation confers high level of ganciclovir resistance, how can the biologist be sure that the UL54 gene is not mutated in the HCMV strain analyzed. In other words if using the NeuMoDx test the result is negative or undetermined could it be due to a mutation in the UL54 gene? To note that UL54 mutation also confers cross-resistance to cidofovir and foscarnet. Please discuss.

We thank the reviewer for this very interesting remark that we had not considered. We contacted Qiagen's research and development department who answered that the target was doubled precisely to compensate for the situation where one of the two targets is not amplified. The result is still positive with 1/2 target amplified. However, the precise sequences of the primers used are protected by a commercial patent. We therefore do not have access to them and an advanced biological exploration is impossible for the moment. However, as this point should not be underestimated, we have added precision in the “materials and methods” section lines 74 “The result is positive even if only one of the two target genes is amplified.”  and a few lines to the discussion as suggested lines 213-218: “It is also noted that the UL54 gene which is one of the targets for amplification by NeuMoDx is a gene which may have mutations that can confer resistance to ganciclovir, cidofovir and/or foscavir [new reference 16]. As the exact sequences of the primers used by NeuMoDx are patent protected we do not know whether they target a conserved sequence or whether resistance mutations may impact on the viral load measurement. However, the dual targeting of the machine limits the risk of false negatives in this case.”

Minor points:

HCMV is latent in bone marrow progenitors, but also in monocytes. This should be corrected accordingly in the text.

Thank you for this precision, we have added it on line 35.

The English has to be checked.

The manuscript was proofread by a native English speaker.

Reviewer 2 Report

The manuscript by Luciani et al. evaluates the NeuMoDx automated system regarding CMV load measurement. Thorough comparison and sequential patient testing makes this evaluation highly valuable for clinical microbiologists, who need real-life feedback on new systems to decide whether to use them.

Minor remarks and questions :

I would like details on turnaround time (maybe a workflow description with timeframes ?), and to know if the 4 hours correspond to the time on-instrument or including centrifugation and sample preparation. Can TAT be longer if there is a significant number of samples being analysed by the system ?

What is the inhibition/invalid rate ?

Correlation analysis would be useful as well as a plot showing the mean bias for samples positive in both techniques (line 91 it is said that all non-zero measurements for both methods were plotted but lines 97 and 99 describes null measurements for one technique or the other).

Details on day hospitals units especially regarding the transplanted organ type would be interesting, especially in terms of viral load threshold for treatment.

I do not really agree that low positives have to be answered as negative to avoid the risk of « over-treatment » because it would be quite rare for a treatment to be initiated without any control sample for such a low viral load ; at least the viral load would have to be controlled and treatment would be decided on this second value. I would instead agree with the sentence stating that increased sensitivity can be really interesting for the early detection of relapses. This aspect should be emphasized in the discussion.

Figure 2 and 3 data are interesting but viral load values would help to better understand the variations in percentage and to visualize the difference in viral load between the two techniques (for Figure 3, the same type of figure with categories of inital viral load value rather than delay would add further information). Testing samples with a third technique would be interesting for patient 9 for which kinetics are drastically opposed. Was there a control sample before treatment was initiated ?

Author Response

The manuscript by Luciani et al. evaluates the NeuMoDx automated system regarding CMV load measurement. Thorough comparison and sequential patient testing makes this evaluation highly valuable for clinical microbiologists, who need real-life feedback on new systems to decide whether to use them.

Thank you for the positive feedback, indeed we think this study could be useful for the medical biological community. 

Minor remarks and questions :

I would like details on turnaround time (maybe a workflow description with timeframes ?), and to know if the 4 hours correspond to the time on-instrument or including centrifugation and sample preparation. Can TAT be longer if there is a significant number of samples being analysed by the system ?

The technical time of the analysis for one sample is shorter. Centrifugation takes 10 minutes and the NeuModx™ takes approximately 1 hour and 10 minutes to perform the viral load measurement (including extraction and PCR). The number of samples that can be processed at the same time depends on the number of modules the machine is equipped with, but the processing time is optimised by grouping the samples by 10 or 12 in order to fill the modules. In our routine, the machine has two modules, with a dedicated technician properly trained on the technical subtleties and anticipating the change of reagents and consumables, we can perform between 40 and 60 CMV viral loads per day with an average delay of 4 hours per sample (including recording, centrifugation, technical execution and biological validation). This has been specified in the text lines 219-225: “The principal advantage of the NeuMoDx™ resides in its "random access" operating mode directly using the primary tube and providing an interpreted result within less than 4 hours based on a frequency of 40 to 60 samplings per day. The strict technical time of execution of the machine for a sample is even shorter: 1 hour and 10 minutes (extraction and PCR) to which it is necessary to add 10 minutes of preliminary centrifugation.  Hence, samples can be processed as they are received in the laboratory without the need to organize for series that delay the time to result to 12-24 hours (i.e R-GENE® technique).”

What is the inhibition/invalid rate?

We have added a few lines to the discussion to further explore the topic of undetermined samples: “During this study the rate of undetermined samples in NeuMoDx™ was 2.64%. This was mainly due to technical errors during the initial use of the machine. However, with hindsight, we have noticed that the rate of undetermined samples varies depending on the patient population. For transplant patients it is now less than 1%, but for intensive care patients it is close to 10%. We assume that samples from this population are more likely to contain PCR inhibitors such as heparin (further investigations are required). Although dilution of the sample to reduce PCR inhibitors may work for some samples, a second PCR technique such as R-GENE® is still needed.” Lines 205-213

Correlation analysis would be useful as well as a plot showing the mean bias for samples positive in both techniques (line 91 it is said that all non-zero measurements for both methods were plotted but lines 97 and 99 describes null measurements for one technique or the other).

The authors would like to thank the reviewer for this remark. We indeed did forgot to discuss the case where the samples where positive in both techniques. These measurements are the cloud of points on the right side of the Bland-Altman plot (figure 1, panel A), where the bias can be observed. It can be seen that NeuMoDx™ measurements tend to be lower than their R-GENE® counterpart, and that this bias decrease for higher measurement values. We also added a correlation test (Spearman rank) of the log value of the measurement for both techniques. We added in the text lines 100-105:

“The cloud of points on the right part of the graph (for mean values above 100, that is for abscise above 2) is the ensemble of measurements where both techniques were positive. We observe in this case that NeuMoDx™ tends to provide lower values than GENE®, with an absolute bias decreasing with the increase of the measurement values. Spearman rank correlation of the log10 measurements for samples positive in both techniques is 0.79 (p < 0.001).”

Details on day hospitals units especially regarding the transplanted organ type would be interesting, especially in terms of viral load threshold for treatment.

Our laboratory works mainly with kidney, heart, liver and lung transplant services. However, each department has developed its own protocol for initiating treatment against CMV reactivation. This depends on a range of biological and clinical arguments. There is no consensus threshold.

I do not really agree that low positives have to be answered as negative to avoid the risk of « over-treatment » because it would be quite rare for a treatment to be initiated without any control sample for such a low viral load ; at least the viral load would have to be controlled and treatment would be decided on this second value. I would instead agree with the sentence stating that increased sensitivity can be really interesting for the early detection of relapses. This aspect should be emphasized in the discussion.

Thank you for this comment, we agree, we have followed your suggestion and modified the text lines 179 - 180: “We therefore assume that is a proper compromise to mitigate the on-off effects likely due to physiological variations while maintaining improved sensitivity comparing to R-GENE®. Additional studies are needed to measure the medical importance of an increased sensitivity during patient follow-up with a special interest for the early detection of relapses.”

Figure 2 and 3 data are interesting but viral load values would help to better understand the variations in percentage and to visualize the difference in viral load between the two techniques (for Figure 3, the same type of figure with categories of initial viral load value rather than delay would add further information). Testing samples with a third technique would be interesting for patient 9 for which kinetics are drastically opposed. Was there a control sample before treatment was initiated?

The authors would like to thank the reviewer for this remark, however we have chosen to present and interpret all the data in percentage of initial load precisely because it was difficult to visualize the variations in absolute values. However we propose to make the data accessible by providing a supplementary table with all the measurements of viral loads in the two techniques for the 15 patients followed. In view of the very favourable evolution of the patient 9 and the non-impact of the NeuMoDx™ on his management, we think that the interest of a third measurement is limited, especially as we do not have a third quantitative technique, only qualitative.
